# Green Finance Policy and ESG Performance: Evidence from Chinese Manufacturing Firms

Xiuli Sun [1,*], Cui Zhou [1] and Zhuojiong Gan [2]

1    School of Statistics, Southwestern University of Finance and Economics, Chengdu 611130, China
2    Department of Economics, Xi'an Jiaotong-Liverpool University, Suzhou 215123, China
*    Correspondence: sunxl@swufe.edu.cn

**Abstract:** While the literature has examined the key role of green finance policy on firms' green innovation and environmental performance, little attention has been paid to firms' environmental, social, and governance (ESG) performance, which is increasingly important to stakeholders. Exploiting heterogeneity in firms' exposure to the green finance pilot zones policy in China in 2017 as a quasi-natural experiment, this paper employs the difference-in-differences model to explore the effect of green finance policy on firms' ESG performance. Based on the data of listed manufacturing firms in China during 2013–2020, our results indicate that the green finance policy could promote firms' ESG performance. Moreover, the overall positive effect is driven mainly by the environmental pillar. Utilizing subsample estimation and the triple differences method, we further find that the higher ESG performance is driven by firms with less financial constraints, firms in economically more developed pilot zones, and state-owned enterprises (SOEs). Mechanism analysis indicates that the pilot policy promotes firms' ESG performance even if it worsens firms' financial constraints. Our study contributes to the research on both the impacts of green finance policy and the relationship between financial constraints and ESG performance, as well as to the literature on ESG structure.

**Keywords:** green finance; ESG; CSR; DID; financial constraints; triple differences

## 1. Introduction

Technological change has greatly raised the living standards of humans. Clearly, humans have been successful in utilizing new technologies. However, we are meeting great challenges in addressing environmental damage and climate change [1]. During the period 2000–2019, there were 7348 massive disaster events resulting from extreme weather around the world, leading to 1.23 million deaths and USD 2.97 trillion in economic losses [2]. World leaders have come to a general consensus on environmental protection. The 2021 United Nations Framework Convention on Climate Change 26th Conference of the Parties (COP26) promised to increase funds for developing countries in order to tackle climate change.

As an economically fast-growing developing country, China accounts for a third of the world's greenhouse gases and 27% of global carbon dioxide [3]. Meanwhile, China is making a great effort to transition to a greener economy. To reach peak carbon emissions before 2030 and achieve carbon neutrality by 2060, China will require a massive shift in resources and new technologies to reduce pollution emissions and enhance energy efficiency and resource productivity. However, limited financial resources might largely hinder the investment of firms in green technology [4]. According to the World Economic Forum's estimation, China needs to close an annual funding gap of about RMB 1.1 trillion (USD 170 billion) [5]. Green finance is designed to make sustainability a part of every firm's financial decision making, and thus, it is promoted by governments and financial institutions, including central banks [6] worldwide in their effort to move towards sustainable development. There is also a continuously increasing interest among academics to evaluate the effectiveness of green finance in promoting green transition [4,7,8].

Because of the existence of externalities, firms are inclined to make excessive investments in polluting industries and inadequate investments in green projects. Economically, the aim of green finance policy is to use policy and institutional arrangements to address the positive externalities of green investments or negative externalities of polluting investments, which cannot be internalized solely by the market [9].

As the green finance system develops and its institutional structure improves, firms are under pressure to disclose environmental, social, and governance (ESG) performance. Firms are being evaluated by sustainability rating agencies such as Thomson Reuters, MSCI, and Bloomberg. ESG ratings are the result of an assessment of a firm's quality, standard, or performance on ESG issues [10]. Sustainable and responsible investors depend strongly on the ESG scores provided by these rating agencies [11]. ESG performance now serves as a critical measurement of corporate sustainability, especially among investors, corporations, and policymakers [12]. Consistent with the increasing importance of ESG performance, more academic studies have tried to identify underlying mechanisms that explain why firms differ in their ESG performance [11,13–16].

ESG means environmentally and socially friendly activities not merely required by law but which go beyond compliance as well, privately supplying public goods, or voluntarily internalizing externalities. Firms adopt different ESG strategies to maximize firm value and minimize costs and risks in the long run. Among all the factors influencing firms' corporate social responsibility (CSR) engagement, Campbell proposed that firms in weak financial situations are less likely to invest in social responsibility [17]. Nevertheless, empirically identifying a causal link running from financial resources to firms' stewardship toward ESG is particularly difficult since unobservable factors may jointly determine a firm's financial resources and its ESG engagement [18].

This paper examines whether green finance affects firms' ESG engagement. To identify, we exploit heterogeneity in firms' exposure to green finance policy in China and pin down the causal impact of green finance on firms' ESG engagement. Green finance seeks to provide financing, operating funds, investments, and other financial services for eco-friendly projects, with ecological preservation as the key driver [19,20]. Whether green finance plays a role in the ESG performance of the corporate sector is still an open question.

Existing literature reveals the close association between green finance activities and sustainable development at both macro and micro levels [20–23]. At the macroeconomic level, for example, based on the data of 25 provinces in China from 2004–2017, Zhang and Wang constructed an evaluation system to assess green finance growth using the Pressure-State-Response model and revealed that green finance could foster sustainable energy development [22]. On the other hand, Liu and Wang provided micro evidence of green finance impacts [20]. Using Chinese listed companies between 2013 and 2020, they demonstrated a robust link between green finance and growing green patent applications.

Green finance studies at the macro level are usually based on the green finance development index, e.g., Zhang and Wang [22]. However, they could easily suffer from endogeneity issues [24]. Policy experiments provide researchers with a good opportunity to explore causality. By examining the impact of green finance policy, scholars have suggested that green finance policy is related to green innovation [8,20,25], corporate investment efficiency [24], controlling the overall situation of air pollution [7], enterprise energy consumption intensity [23], debt-financing cost of heavy-polluting firms [26], and environmental pollution reduction at the macro level [27]. Earlier research focused mainly on the environmental dimension. However, up to now, few studies have explored the combined economic, social, and environmental effects of green finance at the firm level. To the best of our knowledge, the only attempt was made by Li et al., who investigated the effects of green finance policy on firms' ESG performance by examining the 2012 Green Credit Guidelines (GCG) policy in China [28]. They found that the GCG policy enhances ESG performance in firms restricted by the policy, compared to firms without the restrictions.

Our study differs from their analysis in several ways. Firstly, different green finance policies are examined. The GCG policy is designed to force the banking sectors to include environmental requirements in issuing loans to heavy-polluting firms. In other words, the only green financial instrument of the GCG policy is a green loan. On the other hand, the green finance pilot zones policy examined in our paper combines various instruments of green finance such as green bonds, green loans, green insurance, and green fintech. Secondly, Li et al. relied on a self-constructed ESG index while we use Bloomberg ESG as well as MSCI ESG to measure firms' ESG performances [28]. In social responsibility literature, conclusions are sensitive to ESG measurement [29]. The widely used Bloomberg and MSCI ESG indicators allow us to make comparisons with other ESG studies. Thirdly, since the pilot zones policy was implemented in only eight zones, the policy provides us with a natural treatment group and a control group, and thus we do not need to define them ourselves, as was performed by Li et al. [28]. Fourthly, compared with Li et al. [28], we pay particular attention to state-owned enterprises (SOEs). In China, SOEs could typically benefit from soft budget constraints. In fact, Yu et al. showed that green finance policies have primarily eased the financing constraints of SOEs, and green credits are possibly less accessible to privately owned firms (POEs) [4]. Thus, the analysis of SOEs could deepen our understanding of the effects of green finance policy.

We focus on the manufacturing firms in China since the sector has played a key role in China's fast economic expansion, and its value added contributes to about 27.44% of GDP in 2021, according to the World Bank. China serves as a perfect setting for our study. As the world's largest manufacturing power, the manufacturing industry has significant environmental impacts in China. According to China Energy Statistical Yearbook 2021, the manufacturing industry accounts for 57.9% of China's total energy consumption and produces more than 50% of total $CO_2$ emissions. In 2019, the Chinese manufacturing industry contributed 12.24% of the world's carbon dioxide releases and 13.46% of energy consumption [30]. As the central pillar of the Chinese economy, the technological upgrading of the manufacturing sector is a vital strategic task for economic development [31].

By examining listed firms in China in 2013–2020, we study how the green finance pilot zones policy affects firms' ESG performance using the difference-in-differences (DID) estimation. First, the main result is that the pilot zones policy significantly increases the ESG performances of firms. Specifically, for firms exposed to the pilot policy, the ESG score will increase by 7.3% when evaluating the sample mean. Secondly, the overall positive effect of the green finance policy on ESG performance is driven mainly by the environmental pillar rather than the social and governance pillars. Thirdly, utilizing both subsample estimation and triple differences, we further find that the higher ESG performance is mainly driven by heavy-polluting firms, firms with less financial constraints, firms in economically more developed pilot zones, and state-owned enterprises (SOEs). Fourthly, mechanism analysis indicates that the pilot policy promotes the ESG performances of firms even if it worsens firms' overall financial constraints. The reason for this might be that the policy aims at environment-friendly projects. Finally, the results are robust to the parallel trend test, PSM-DID, alternative ESG proxy, and placebo test.

Our paper sheds light on the role of green finance in firms' sustainable performance. Scholars have noticed the key role of green finance in promoting the diffusion of environmental innovation [32], carbon mitigation [19], environmental responsibility [33], and green performance and innovation [34]. Much of the green finance literature has focused on environmental performance, but little attention has been paid to ESG performance, which is increasingly important to stakeholders [10,11]. Utilizing the Bloomberg ESG database and the data of listed manufacturing firms in China from 2013 to 2020, our results indicate that the green finance policy could promote the ESG performances of firms.

Furthermore, our findings contribute to research examining the role of financial constraints for CSR. Leong and Yang confirmed the negative effects of financial constraints on all dimensions of CSR performance [35]. Hong et al. showed that financial constraints are a critical obstacle to corporate social responsibility [18]. They showed that when firms'

constraints are exogenously relaxed, firms with higher financial constraints improve their CSR performance relative to less-constrained firms. Xu and Kim found that the relaxation of financial constraints reduces US-listed firms' toxic emissions [36]. We find that the pilot policy tightens firms' financial constraints but promotes their ESG performance. The reason for this might be that the policy is targeted to improve firms' environmental performance. More importantly, the overall positive effect of the green finance policy on ESG performance is driven mainly by the environmental pillar rather than the social and governance pillars.

The paper makes an additional contribution by shedding light on the role of green finance policy on sustainable development. Previous literature has related the same pilot zones policy to environmental pollution control [27], high-quality green innovation [37], the decrease of debt-financing cost [26], the increase of green patent output [20], the reduction of inefficient and excessive investments [24], environmental quality improvement [2], overall air pollution control [7], and the reduction of energy consumption intensity [23]. Lu et al. investigated the Green Credit Guidelines in China in 2012 and found that the green finance policy increases the financial constraints and debt financing cost of high-polluting enterprises. Li et al. examined the same Green Credit Guidelines (GCG) policy. Their results showed that the green finance policy promotes restricted firms' social responsibility even though it tightens financial constraints. Restricted firms are firms restricted by GCG, and they are heavily polluting firms. We reach a similar conclusion to Li et al. by examining the green finance pilot policy in 2017. We further point out that the higher ESG performance is mainly driven by heavy-polluting firms, firms with less financial constraints, firms located in economically more developed pilot zones, and SOEs.

The remainder of this paper is organized as follows. Section 2 presents the institutional background and provides the literature review. Section 3 provides an empirical strategy. Section 4 describes the data source and descriptive statistics. Section 5 provides the empirical results, including parallel trend tests, heterogeneity analysis, and mechanism analysis. Section 6 describes the robustness tests. Section 7 provides a conclusion.

## 2. Related Literature

### 2.1. Institutional Background

After decades of fast economic growth, China has now prioritized developing an environmentally friendly economy. In September 2020, President Xi promised that China would reach the maxima of carbon emissions by 2030 and net-zero of all six types of greenhouse gas releases and carbon neutrality by 2060 [16]. Achieving these goals involves substantial economic and social transformation and immense investment. This will necessitate a strong financial system to stimulate, facilitate, and steer finance into these areas [38].

Green finance is specified by the People's Bank of China (PBOC) as "financial services that support economic activities aimed at improving the environment, mitigating climate change, and utilizing resources more efficiently" [39]. In other words, any structured financial activity, a product or service, which has been created to stimulate the development of green projects, minimize the environmental damage of regular projects, or a combination of both, could be viewed as green finance.

China formulated the green finance development plan as early as the 1990s and has strived to develop the green finance system over the years. Green finance offers an economic means to bridge the gap between the supply and demand of sustainable economic development [7]. Though still at an early stage, China has made impressive progress in improving the green financial system and markets. China's green bond market has emerged as the world's second-largest, after the US [40]. According to PBOC, China issued 1643 green bonds until the end of the year 2021, with a total balance of RMN 1727 billion (USD 270 billion) [41].

The green finance pilot zones are part of the central government's efforts to establish a strong green financial system. The first batch was initiated in June 2017 in eight regions across eastern, central, and western parts of China, at different development stages [24]. The aim of establishing the pilot zones was to gain experience to think through scale issues

as policies are elevated nationally. Specifically, as indicated in Figure 1, they are Huzhou city and Quzhou city in Zhejiang province, Ganjiang New District (located in Nanchang city) in Jiangxi province, Guangzhou city in Guangdong province, Gui'an New District (located in Guiyang and Anshun) in Guizhou, and Hami city, Changji prefecture, and Kramay city in Xinjiang.

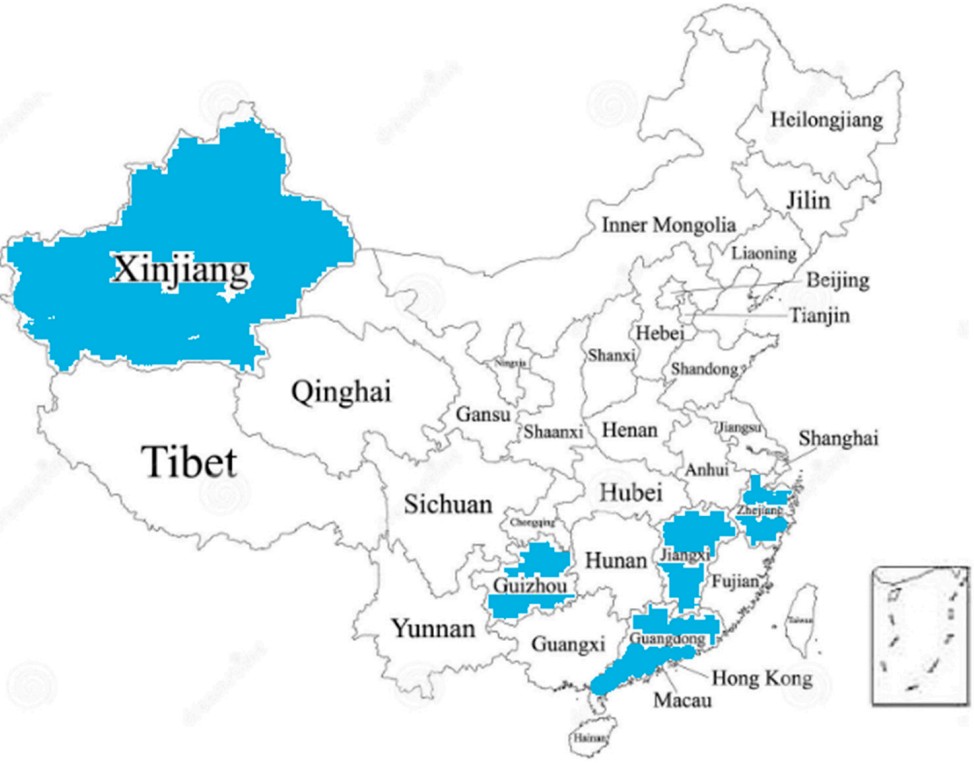

**Figure 1.** Eight green finance pilot zones in five provinces in China. In June 2017, eight regions in Zhejiang, Guangdong, Jiangxi, Guizhou, and Xinjiang, were selected as the first batch of pilot zones for green finance reform.

The primary goals for the pilot zones are to facilitate the establishment of green finance departments or branches within financial institutions, promote green credits, establish markets for trading rights over emission, energy use, and water, develop public service channels for green industries, and build risk management mechanisms for green finance [7]. However, the focuses of the eight regions in the five provinces differed [26], as shown in Table 1, since they were at different stages of development. For example, the Guangzhou zone aimed to develop financial products and services innovation, green fintech, and green finance markets in cooperation with Hongkong and Macao capital markets. At the same time, the Xinjiang zone focused mainly on developing green finance for agriculture modernization, clean energy, and high-end manufacturing related to renewable energy.

The pilot zones have achieved significant milestones. As of 2020, the outstanding green loans in the pilot zones reached RMB 236.8bn (USD 35bn), accounting for 15.1% of the overall lending portfolio [42]. The Green Credit Platform developed in Huzhou has assisted over 17,000 enterprises to obtain more than ¥187 billion in financing as of July 2020 [43]. Guangzhou has a robust carbon pilot trading market, the first green project financing integration system, and real momentum in coordinating efforts to stand up green finance in the Greater Bay Area. In fact, according to Lv et al., the green finance development of these five provinces was ranked top six, with Beijing ranking first [44].

**Table 1.** Priorities in pilot zones.

| Province | Municipality | Priorities |
|---|---|---|
| Zhejiang | Huzhou city<br>Quzhou city | 1. Innovating green finance to foster the transformation of traditional industries, including green credit systems, green bank rating, green insurance, etc.<br>2. Green financing for micro, small, and medium enterprises |
| Guangdong | Guangzhou city | 1. Green financial products and services innovation, including establishing environmental equity trading markets, green bonds, green insurance, etc.<br>2. Developing green fintech<br>3. Developing green finance markets in cooperation with Hongkong and Macao capital market |
| Jiangxi | Ganjiang New District (located in Nanchang city) | 1. Establishing fully operational green finance system with numerous products and services<br>2. Upgrading traditional industries |
| Guizhou | Gui'an New District (located in Guiyang and Anshun cities) | 1. Optimizing local ecological resources in developing green finance to boost clean and sustainable economic growth<br>2. Alleviating poverty<br>3. Building sharing infrastructure for big data |
| Xinjiang | Hami City, Changji Prefecture, and Karamay City | 1. Demonstrating role in the Green Belt and Road<br>2. Developing green finance to foster agriculture modernization, clean energy, and high-end manufacturing related to renewable energy |

An additional pilot zone located in Lanzhou New District in Gansu province was announced in 2019. The priority of the Lanzhou zone is to develop a local green finance system and to promote electric vehicles and big data industries. We do not include it in this paper since we only focus on the first batch of pilots.

### 2.2. Related Literature

2.2.1. Factors Influencing Firm ESG Engagement

Existing studies have proposed various theories on why firms invest in ESG. Among them, the shareholder expense view and the stakeholder view are the two main views. The stakeholder view argues that CSR activities could increase shareholder wealth since the satisfaction of other stakeholders' interests makes them more supportive of the firm, which contributes to maximizing shareholder wealth. That is, stakeholder theory argues that corporate managers have to meet the need of stakeholders to maximize firm value [45,46]. Contrary to the stakeholder view, the shareholder view argues that managers' engagement in social responsibility benefits other stakeholders at the cost of shareholders. In other words, the shareholder view argues that CSR damages shareholder value [47–49].

To legitimize ESG on sound economic grounds, numerous studies have investigated the effects of ESG on firms. Prior literature has documented that ESG is positively related to firm financial performance [50–53]. Prior studies also related ESG to different aspects of firms, such as firm value [54], risk mitigation [55], insurance mechanism [56], and access to finance [57].

ESG is a strategic decision of the firm. Scholars have investigated the factors that influence firms' ESG engagement. Firm characteristics, such as ownership structure [58] and firm size [16,59–61], have been proven to be vital in firms' ESG performances in numerous studies. Academics have also started to focus on the role of corporate governance in ESG, including board structure [60,62], CEO duality [60,63], female board members [64,65], and the presence of independent directors [16,60,61,66]. The underlying philosophy is that since managers could significantly affect a firm's major decisions, they should also have a critical influence on its ESG engagement decisions. We briefly summarize the related literature on ESG performance in Table 2.

**Table 2.** Literature summary.

| Paper | Theory | ESG Indicators | Main Findings |
|---|---|---|---|
| Baldini et al. (2018) [14] | Stakeholder theory | Bloomberg ESG; firms from Japan, the US, the UK, and Australia during the 2005–2012 period | A country's cultural system, labor system, and political system could largely affect firms' ESG disclosure. |
| Chan et al. (2017) [13] | Shareholder theory | MSCI ESG STAT ratings; firms listed in the US market from 1992 to 2010 | The results suggested that financial constraints are negatively linked to CSR activities. |
| DasGupta (2022) [15] | Stakeholder theory | Thomson Reuter's Asset4 ESG ratings; firms from 27 countries during 2010–2019 | The paper found that financial performance shortfall could positively affect a firm's ESG performance. |
| Drempetic et al. (2020) [11] | Stakeholder theory | Thomson Reuters ASSET4 ESG ratings; firms from the USA, Japan, Great Britain, and Canada during the 2004–2015 period | The ESG scores are biased towards bigger firms due to the necessity of resources for obtaining ESG data. |
| Galbreath (2016) [66] | Shareholder theory and stakeholder theory | Sustainable Investment Research Institute (SIRIS) CSR ratings; firms in the Australian Securities Exchange 300 (ASX300) from 2004–2009 | External directors and female board members complement each other, as their interaction effect gradually affects CSR beyond their individual, independent effects. |
| Hansen et al. (2018) [49] | Shareholder theory and stakeholder theory | MSCI ESG KLD STATS; US firms in the manufacturing sector for 1992–2009 | The results show that CSR investment and its economic influences are context specific. |
| Liao et al. (2018) [62] | Shareholder theory and stakeholder theory | Rankins CSR ratings; listed firms over the period of 2008–2012 in China | Larger board sizes, separation of CEO and chairman positions, and more female directors are positively linked to CSR engagement. |
| Oh et al. (2011) [58] | Shareholder theory | KEJI index; listed Korean firms in 2005 | Institutional and foreign investors tend to engage CSR more. |
| Shu and Tan (2023) [16] | Stakeholder theory | Hexun ESG; listed firms in China during the 2010–2019 period | Carbon control policy risk reduces firms' ESG engagement, with bank loan costs and financial constraints as underlying mechanisms. |
| Yuan et al. (2019) [63] | Shareholder theory and stakeholder theory | MSCI STATS CSR ratings; US firms from 2003 to 2012 | CEO ability enhances firms' CSR ratings. |

However, only a few studies of ESG noticed the effect of financial resources on a firm [18,67]. Based on KLD CSR ratings and the dataset of S&P 500 firms from 1991 to 2008, Hong et al. pointed out that neglecting the heterogeneity of firms' financial constraints could bring in a spurious link between financial performance and corporate social responsibility, even if the motives for social responsibility are not-for-profit [18]. In their main identification strategy, they used a difference-indifferences method by exploiting the exogenous shock of the Internet bubble of the late 1990s. Based on MSCI ESG STAT ratings and the data of listed firms in the US market from 1992 to 2010, Chan et al. investigated the association between cash flow liquidity and CSR, and they confirmed a significant negative relationship between financial constraints and CSR activities [13]. However, they fail to establish a causal relationship that less financially constrained firms engage more in CSR. Exploiting exogenous variation in financial constraints, Hong et al. showed that financial constraints are a critical driver of CSR [18]. In spite of the motives for CSR engagement, less constrained firms have a larger expenditure on social responsibility.

ESG spans three dimensions of a firm's activities and captures a firm's efforts to address various externalities that it produces while pursuing profit maximization that is not internalized by shareholders [68]. Thus, beyond viewing ESG as a whole, we consider different dimensions of ESG as fundamental tradeoffs among different stakeholders. The engagement activities across different ESG dimensions represent various aspects of stakeholder concerns [68]. According to the stakeholder theory, allocating firm resources

to social responsibility not directly relevant to primary stakeholders may not result in sustainable competitive advantage [69,70]. The relationship between CSR engagement and firm financial performance varies with CSR dimensions. For example, using the MSCI ESG database and annual data from COMPUSTAT from 1991 to 2012, Tsai and Wu showed that the practices of social responsibility pertaining to human rights and the environment matter primarily during a crisis period when the financial resource is finite [71]. On the other hand, firms in a financial crisis may reduce their engagement in employee relations to enhance their financial performance. Benlemlih and Bitar differentiated between primary and secondary stakeholders [72]. Based on the data of US firms during 1998–2012, their findings indicated that CSR components directly concerning firms' primary stakeholders are more critical in improving investment efficiency than those linked to secondary stakeholders. Based on a sample of US-listed firms from 1991 to 2013, Bouslah et al. suggested that firms specialized in CSR dimensions, with about three-quarters of them centering on a single aspect [73]. The degree of specification varies with industries, and the focused dimension also differs even within industries with similar specification levels.

2.2.2. Green Finance and ESG

Long-term and stable capital investment is a prerequisite for firms to conduct green production [4]. Thus, a firm's green development is limited by financial resources. A lack of financial resources could largely hinder investment in green technology for two reasons [4,74]. Moreover, compared to regular innovation, green technology transition is characterized by "double externalities" as it generates knowledge spillovers and environmental spillovers at the same time [25,75]. To alleviate financial constraints, public policies are needed to promote green economy transformation [76]. Among different public policies, green finance is a key financing tool to address environmental damage.

As to the mechanisms underlying the link between green finance and firms' environmental performance, scholars mainly focus on financing costs or financial constraints. Liu et al. established a financial computable general equilibrium model (CGE) [77]. The simulation results showed that with the implementation of the green credit policy, the average financing costs of the energy-intensive industries increase by 1.1% in the short run, decrease by 0.008% in the medium run, and increase by 0.003% in the long run [77]. By investigating the panel data of 52 green firms and 81 "two-high" (high-pollution and high-emissions) firms in China from 2001–2007, Xu and Li concluded that green finance has asymmetric impacts on heavy-polluting firms and relatively clean firms. Specifically, the green credit policy raises the debt financing cost of "two-high" firms but lowers that of green firms [78].

## 3. Empirical Models and Estimation Strategies

### 3.1. Baseline Regressions

To study the effect of the pilot zones policy on firms' ESG performance, we utilize a difference-in-difference (DID) strategy. We take advantage of the fact that the green finance pilot zones policy is an exogenously applied policy for all firms since it is a top-down policy. Huang et al. argued that pilot zones are decided by the central government rather than the local government, which can mitigate the endogeneity concern arising from the self-selection bias [37]. Thus, the policy can be considered a quasi-natural experiment [26]. We then specify our empirical specification as:

$$ESG_{it} = \beta_0 + \beta_1 treat_i + \beta_2 post_t + \beta_3 treat_i \times post_t + \theta X_{it} + \lambda_i + \mu_t + \varepsilon_{it}, \quad (1)$$

where $ESG_{it}$ represents the firm $i$'s ESG performance in year $t$. $treat_i$ is a dummy variable that is equal to 1 if the firm is located in the pilot city and is otherwise 0. $post_t$ indicates whether or not the policy was implemented, and equals 1 for 2017–2020 and 0 otherwise. $treat_i \times post_t$ is the interaction of $treat_i$ and $post_t$ variables, and the coefficient $\beta_3$ is the main interest of the study, which captures the causal effect of green finance policy on firms' ESG performance. Moreover, $X_{it}$ is a vector of all the control variables, including

firm age, firm size, ROA, duality, female directors, independent directors, the share of the largest shareholder, SOE, leverage, growth, free cash flow, and financial constraint indicator. Firm size is measured as the log form of total assets. To capture the effects of macroeconomic factors, we also include the log form of city-level GDP per capita in the regression. Individual firm fixed effects, province fixed effects, industry fixed effects, and year fixed effects are also controlled in our specification.

Cash generally represents a substantial share of a firm's financial resources, allowing firms to make investments quickly. Thus, cash holding facilitates firms to avoid asymmetric information costs and transaction costs related to external financing.

The treatment and control groups need to follow parallel trends before the policy implementation to ensure the validity of the DID method. To test the validity of the parallel trends assumption, following the literature [79,80], we estimate the annual policy effects, which also capture the dynamics of the policy. We then estimate the following model:

$$ESG_{it} = \beta_0 + \sum_{t=2014}^{2020} \gamma_t \left( treat_i \times Year_t \right) + \theta X_{it} + \lambda_i + \mu_t + \varepsilon_{it}, \tag{2}$$

where year dummies, $Year_t$, are interacted with the treatment indicator, $treat_i$, to generate a DID estimate for each year, using the year 2013 as a benchmark. We expect the post-policy yearly effects to be significant and positive and the pre-policy yearly effects to be insignificant.

### 3.2. Triple Differences

Environmental damage is mainly caused by heavy-polluting firms. More specifically, the heavy-polluting sector is the main emitter of pollutants like nitrogen oxides (NOx), sulfur dioxide ($SO_2$), and particulate matter (PM), which contribute about 99%, 98%, and 97% [81]. Therefore, investigation of the effects of green policy on heavy-polluting firms is particularly vital. As the prime targets of environmental regulation, the policy effects might be even more significant among heavy-polluting firms relative to non-heavy-polluting firms.

To directly test if heavy-polluting firms have a larger effect, we implement a triple differences test by separating both the treatment and control sample into heavy-polluting and non-heavy-polluting firms, defined based on heavy-polluting and non-heavy industries as in Shi et al. [26]. Our *pollution* indicator will not vary through the sample period.

Following previous studies [79,82,83], we explain the outcome variable $ESG_{it}$ with a treatment dummy ($treat_i$), a post-event dummy variable ($post_t$), a heavy-polluting dummy ($pollution_j$), the whole set of double interaction terms ($treat_i \times pollution_j$, $pollution_j \times post_t$, and $treat_i \times post_t$), and the triple interaction term ($treat_i \times pollution_j \times post_t$), controlling for a set of fixed effects:

$$\begin{aligned} ESG_{it} = \alpha \times treat_i \quad & \times pollution_j \times post_t + \beta_1 \times treat_i \times pollution_j \\ & + \beta_2 \times pollution_j \times post_t + \beta_3 \times treat_i \times post_t \\ & + \beta_4 X_{it} + \gamma_i + \varepsilon_{it}. \end{aligned} \tag{3}$$

The specification includes all three double interaction terms to operationalize the triple differences estimator. Following the previous literature, we also include firm fixed effects ($\gamma_i$) to capture firm-level time-invariant heterogeneity [84]. The critical coefficient of interest is the one on the triple interaction term, $\alpha$. It estimates whether the difference between the differential response of the heavy-polluting treated firms relative to their heavy-polluting control group and the differential response of the non-heavy-polluting treated firms relative to their non-heavy-polluting control group is significant after the implementation of the green finance policy.

## 4. Data

Our data on ESG were obtained from the Bloomberg database. The scores, ranging from 0.1 to 100, measure the transparency or disclosure quality, for a broad range of ESG dimensions, such as pollution waste disposal, greenhouse gas emissions, renewable energy, community relations, diversity, human rights, political donations, executive compensation, board size, independent directors, and employee turnover. The Bloomberg ESG scores summarized these aspects in three dimensions, environmental, social, and governance pillars, each with a 33% weighting [60]. Moreover, Bloomberg provides scores for each pillar, i.e., environmental, social, and governance scores.

One of the most broadly used ESG scores by institutional investors is provided by MSCI, formally known as KLD Research and Analytics. The MSCI score comprehensively evaluates each firm's ESG profile [71]. The Bloomberg ESG indicator is also extensively used in ESG studies in China [61,85–87] and in other countries [10]. Bloomberg data are more consistent than MSCI, and thus we chose Bloomberg ESG as our data source [86].

Compared to the MSCI data, the Bloomberg ESG is adjusted to different industries. Therefore, a firm is assessed using information relevant to its industry [14]. We use both the composite ESG score and the three component scores in our analysis.

We obtain firm characteristics and financial performance data from the China Stock Market and Accounting Research Database (CSMAR) to supplement our analysis. Our dataset consists of all manufacturing firms listed on the stock exchanges in Shanghai and Shenzhen between 2013 and 2020. We matched the ESG data and firm characteristics and financial performance data from CSMAR, excluding firms with special treatment (ST) type and missing data.

Table 3 summarizes descriptive statistics for the sample. The average ESG score is 21.39. The average environmental, social, and governance scores are 11.85, 23.59, and 44.43, respectively. Our ESG scores are consistent with other studies [61]. The average firm age is 18.59, ranging from four to 39. Besides, the mean value of SOE is 0.45, indicating that 45% of the sample are state-owned enterprises. The average firm size is 22.95. The average firms are moderately levered with a leverage ratio of 45%, a mean return on assets (ROA) of 0.04, and a mean share of the largest shareholder of 35.74%. The average growth rate, measured as the sales revenue growth rate, is 22%. The average number of female directors and the average number of independent directors are 2.83 and 3.27, respectively. The average CEO duality is 0.23, indicating that 23% of CEOs are the board's chairman at the same time. An average firm has a mean free cash flow level of 0.09. Furthermore, the statistics of the variables demonstrate that substantial variance exists among samples.

We use the Size-Age (SA) index proposed by Hadlock and Pierce to measure financial constraints [88]. The SA index is calculated using firm size and age, where firm size is measured by the natural logarithm of total assets and firm age by the total number of years since a firm was established. The SA index is computed as follows:

$$SA = -0.737 \times Size_{it} + 0.043 \times Size_{it}^2 - 0.040 \times Age_{it}. \tag{4}$$

The average of the SA index is $-3.83$, ranging from $-4.69$ to $-2.76$. The correlation between the SA index and firm size in our sample is 0.2193. Notice that the SA index is convex in firm size, and it increases with firm size when the firm size is larger than 8.57. In our sample, the average firm size is 22.95, ranging from 19.55 to 27.55. Consistent with the previous studies, a larger SA index (i.e., smaller absolute value) indicates a less severe financial constraint [89–91].

As indicated by Wang et al., the development of ESG ratings in China is still in the early stage, and only about 30% of listed firms have ESG ratings on average [85]. Table 4 presents the distribution of observations across the industries. Following Shi et al. [26], we define firms in the following industries as heavy-polluting firms as indicated in Table 4. The heavy-polluting industries account for 50.3% of all the observations. Firms in other industries are classified as non-heavy-polluting firms. Table 4 also shows the average ESG

scores across different industries. Waste resources and material recovery and processing (C42) has the highest ESG score, 33.28, while another manufacturing industry (C41) has the lowest ESG. Among all the heavy-polluting industries, ferrous metal smelting and rolling pressing (C31) has the highest ESG (25.88), while rubber and plastic products (C29) has the lowest ESG (19.03). Moreover, the ferrous metal smelting and rolling pressing industry scores the highest in both the environmental dimension (E) and the social dimension (S), with scores of 16.85 and 26.95, respectively. Regarding the governance pillar, the automobile manufacturing industry (C36) scores the highest, with a score of 46.61.

**Table 3.** Summary statistics.

| Variables | Obs | Source | Mean | Std. dev. | Min | Max |
|---|---|---|---|---|---|---|
| ESG | 4486 | WIND | 21.39 | 7.00 | 1.24 | 64.11 |
| Environmental dimension | 4047 | CSMAR | 11.85 | 8.54 | 0.78 | 65.63 |
| Social dimension | 4473 | CSMAR | 23.59 | 9.26 | 3.51 | 77.19 |
| Governance dimension | 4486 | CSMAR | 44.43 | 5.19 | 3.57 | 64.54 |
| Log(asset) | 4486 | CSMAR | 22.95 | 1.21 | 19.55 | 27.55 |
| Age | 4486 | CSMAR | 18.59 | 5.31 | 4 | 39 |
| ROA | 4486 | CSMAR | 0.04 | 0.12 | −3.91 | 0.37 |
| Duality | 4486 | CSMAR | 0.23 | 0.42 | 0 | 1 |
| Number of female directors | 4486 | CSMAR | 2.83 | 1.72 | 0 | 10 |
| Number of independent directors | 4486 | CSMAR | 3.27 | 0.65 | 2 | 8 |
| Largest shareholder proportion | 4486 | CSMAR | 35.74 | 14.99 | 3.39 | 89.99 |
| SOE | 4486 | CSMAR | 0.45 | 0.50 | 0 | 1 |
| Leverage | 4486 | CSMAR | 0.45 | 0.23 | 0.0080 | 8.01 |
| Firm growth | 4486 | CSMAR | 0.22 | 1.82 | −0.98 | 58.84 |
| Free cash flow | 4486 | CSMAR | 0.09 | 1.70 | −24.44 | 39.69 |
| SA | 4486 | CSMAR | −3.83 | 0.24 | −4.69 | −2.76 |
| Log (city-level GDP per capita) | 4194 | China City Statistical Yearbook | 11.38 | 0.49 | 9.54 | 12.28 |
| Province-level GDP per capita growth | 4486 | China City Statistical Yearbook | 0.07 | 0.03 | −0.039 | 0.21 |

The units of total asset and city-level GDP per capita are RMB Yuan.

**Table 4.** Number of firms and ESG over industries (Heavy-polluting industries are in bold).

| Industry Code | Industry Name | Number of Firms | ESG | E | S | G |
|---|---|---|---|---|---|---|
| **C13** | **Agro-food processing** | **110** | **19.80** | **9.54** | **24.04** | **42.47** |
| C14 | Food Manufacturing | 78 | 20.45 | 10.76 | 24.06 | 45.19 |
| **C15** | **Beverage manufacturing** | **193** | **19.83** | **11.34** | **20.04** | **44.48** |
| **C17** | **Textile industry** | **80** | **20.10** | **9.92** | **23.13** | **42.01** |
| C18 | Textile and garment industry | 91 | 16.92 | 6.63 | 20.11 | 43.78 |
| **C19** | **Leather, fur, feather products, and shoe manufacturing** | **16** | **20.04** | **9.69** | **20.18** | **43.75** |
| C21 | Furniture manufacturing | 12 | 18.35 | 7.83 | 21.64 | 42.26 |
| **C22** | **Paper making and paper products** | **78** | **22.07** | **12.46** | **22.33** | **46.52** |
| C23 | Printing and record medium reproduction | 38 | 19.19 | 7.88 | 22.67 | 43.33 |
| C24 | Education and sports goods | 16 | 20.66 | 8.87 | 27.08 | 41.29 |
| **C25** | **Petroleum processing, coking, and nuclear fuel processing industry** | **63** | **20.46** | **10.39** | **21.39** | **46.34** |
| **C26** | **Chemical raw materials and chemical products** | **460** | **21.15** | **11.22** | **23.67** | **44.15** |
| **C27** | **Medical and pharmaceutical products** | **511** | **22.52** | **13.26** | **25.68** | **44.02** |
| **C28** | **Chemical fiber industry** | **65** | **23.62** | **13.42** | **24.29** | **46.43** |
| **C29** | **Rubber and plastic products** | **79** | **19.03** | **7.83** | **22.83** | **42.81** |
| **C30** | **Nonmetallic mineral products** | **185** | **23.18** | **13.99** | **24.50** | **45.06** |
| **C31** | **Ferrous metal smelting and rolling pressing** | **163** | **25.88** | **16.85** | **26.95** | **45.92** |
| **C32** | **Nonferrous metal smelting and rolling processing** | **255** | **23.64** | **13.98** | **25.64** | **45.09** |
| C33 | Metallic mineral products | 90 | 20.19 | 11.09 | 21.33 | 43.00 |
| C34 | General equipment manufacturing | 214 | 20.52 | 10.27 | 24.15 | 43.41 |
| C35 | Special equipment manufacturing | 305 | 20.69 | 10.65 | 23.15 | 44.88 |
| C36 | Automobile manufacturing | 183 | 24.40 | 14.72 | 26.40 | 46.61 |
| C37 | Transportation equipment | 145 | 20.76 | 11.36 | 20.53 | 45.96 |
| C38 | Electrical machinery and equipment manufacturing | 380 | 20.57 | 10.83 | 22.69 | 44.01 |
| C39 | Communications equipment, computers, and other electronic equipment | 583 | 20.80 | 11.70 | 22.83 | 44.39 |
| C40 | Instrument and meter manufacturing | 51 | 18.94 | 8.51 | 21.67 | 43.03 |
| C41 | Other manufacturing | 29 | 15.19 | 8.63 | 14.34 | 41.13 |
| C42 | Waste resources and material recovery and processing | 13 | 33.28 | 26.42 | 39.00 | 43.27 |

## 5. Results

### 5.1. Baseline Regression Results

We empirically examine the effects of green finance by examining the green finance pilot zones policy through the DID model. Green finance might be endogenous since policymakers might make green finance development decisions based on firm performance [39]. Compared to the ordinary least squares (OLS) method, DID can solve endogeneity problems [92]. Similarly, Li et al. investigated the effects of green finance on CSR by examining the 2012 GCG policy through the DID model [28].

Table 5 reports the effect of the green finance pilot zones policy on the ESG performance of a firm. Robust standard errors are used across all specifications. Column (1) examines the basic impact of the green finance policy on the ESG performance of firms.

**Table 5.** Baseline results.

| | (1) | (2) | (3) | (4) | (5) |
|---|---|---|---|---|---|
| Treat * post | 1.538 *** | 1.600 *** | 1.607 *** | 1.595 *** | 1.574 *** |
| | (0.404) | (0.404) | (0.404) | (0.408) | (0.407) |
| Post | 0.537 *** | 0.522 *** | 0.491 *** | 0.488 *** | 2.363 * |
| | (0.172) | (0.180) | (0.180) | (0.180) | (1.436) |
| Firm size | 1.082 *** | 1.082 *** | 1.151 *** | 1.230 *** | 1.217 *** |
| | (0.192) | (0.200) | (0.201) | (0.212) | (0.213) |
| Firm age | 1.270 *** | 1.249 *** | 1.229 *** | 1.249 *** | 0.971 *** |
| | (0.0911) | (0.0945) | (0.0939) | (0.0956) | (0.231) |
| ROA | 0.261 | 0.462 | 0.485 | 0.506 | 0.565 |
| | (0.548) | (0.575) | (0.564) | (0.563) | (0.566) |
| Duality | 0.113 | 0.0934 | 0.0893 | 0.145 | 0.135 |
| | (0.207) | (0.209) | (0.209) | (0.211) | (0.211) |
| Female directors | 0.0711 | 0.126 ** | 0.131 ** | 0.122 ** | 0.121 ** |
| | (0.0596) | (0.0612) | (0.0612) | (0.0614) | (0.0612) |
| Independent directors | −0.550 *** | −0.564 *** | −0.554 *** | −0.524 *** | −0.519 *** |
| | (0.167) | (0.173) | (0.175) | (0.175) | (0.176) |
| Largest shareholder | 0.0366 *** | 0.0349 *** | 0.0350 *** | 0.0376 *** | 0.0387 *** |
| | (0.0109) | (0.0111) | (0.0112) | (0.0117) | (0.0117) |
| SOE | 0.829 ** | 0.816 ** | 0.823 ** | 0.737 * | 0.788 ** |
| | (0.368) | (0.396) | (0.402) | (0.381) | (0.382) |
| Leverage | −0.481 | −0.610 * | −0.637 * | −0.660 * | −0.619 * |
| | (0.317) | (0.349) | (0.358) | (0.373) | (0.370) |
| Growth | −0.0175 | −0.0176 | −0.0276 | −0.00720 | −0.00905 |
| | (0.0243) | (0.0249) | (0.0301) | (0.0274) | (0.0276) |
| Free cash flow | 0.0604 | 0.0411 | 0.0421 | 0.0394 | 0.0409 |
| | (0.0681) | (0.0691) | (0.0696) | (0.0700) | (0.0701) |
| SA | 21.26 *** | 20.61 *** | 20.66 *** | 21.00 *** | 21.12 *** |
| | (2.008) | (2.049) | (2.061) | (2.110) | (2.108) |
| Log(GDP) | | −0.249 | 0.127 | −0.0324 | −0.330 |
| | | (0.537) | (0.535) | (0.542) | (0.562) |
| Constant | 54.05 *** | 54.80 *** | 50.04 *** | 49.72 *** | 57.87 *** |
| | (7.274) | (9.093) | (8.898) | (9.480) | (10.60) |
| Industry dummies | No | No | No | Yes | Yes |
| Province dummies | No | No | Yes | Yes | Yes |
| Year dummies | No | No | No | No | Yes |
| Firm dummies | Yes | Yes | Yes | Yes | Yes |
| $N$ | 4478 | 4186 | 4186 | 4186 | 4186 |
| adj. $R^2$ | 0.795 | 0.798 | 0.798 | 0.799 | 0.799 |

Standard errors in parentheses, * $p < 0.10$, ** $p < 0.05$, *** $p < 0.01$. The standard errors are computed using robust standard errors.

Column (2) includes the log form of city-level GDP per capita to capture the macroeconomic impacts. Column (3) also includes province dummies to control for provincial fixed effects. In addition, industry dummies are added in column (4), and year dummies are further included in column (5). We use the specification of column (5) as our baseline model. The estimate of *Treat * Post* is 1.574, indicating that the implementation of the green finance pilot zones policy has a significant and positive effect on firms' ESG performance. The result is consistent across all specifications. This result corroborates the studies of Li et al., Pang et al., and Su et al. [23,28,93], who found that green finance policy leads to firms' CSR increases and environmental performance.

Consistent with other ESG studies in China, our result is also economically nontrivial. Based on the estimations in column (5), the ESG score increases by 1.57, which is around 7.3% when evaluating at the sample mean of ESG. By examining listed firms in China during 2010–2019, Shu and Tan found that the carbon control policy reduces firm ESG performance by 0.011, which varies by 8.308% when the risk of carbon control policy varies by one standard deviation [16]. Fang et al. examined the impact of digitization on Bloomberg ESG using listed firms in China from 2012 to 2020. Their results suggested that the increase in digitalization could account for about 7.69% of the rise in ESG scores over the sample period.

The coefficients of other control variables are in line with our expectations. The results in column (5) show that larger firm size, older firms, the number of female directors, higher ownership concentration, SOE, and fewer financial constraints are all related to a higher ESG score. Leverage has a negative effect on ESG performance.

Equation (1) identifies the average effect of the green finance policy. Practically, the policy may have a time lag effect because the impact of the pilot zones policy might become more observable three or four years after the implementation. Moreover, the assumption of our DID identification strategy is that the different over-time changes between the treatment group and the control group should come solely from the policy implementation in 2017 and not from any preceding different time trend across firms. To test this assumption, in Equation (2) we replace the treat-post interaction in Equation (1) with the sum of the treat-year interactions as in previous studies [7,25,28,29,38]. The dynamic effects of the policy are presented in Table 6. As in Table 5, we also include different sets of industry, provincial and year dummies to control different fixed effects across different columns, with column (3) of Table 6 including the full set of fixed effects. As presented in column (3) of Table 6, all coefficients of interactions *treat * year* are insignificant before and at the time of policy implementation, 2017, implying insignificant differences in pre-trends, suggesting that the parallel trend assumption of the DID method is not violated. The coefficients of interactions *treat * year* are positive and statistically significant. The policy impact gradually increases after the initial policy year, consistent with our expectation.

The results in column (3) of Table 6 are graphed in Figure 2. The interval for each point indicates the 95% confidence interval. Clearly, there was no significant effect on firms' ESG performance before the implementation of the pilot zones policy in 2017. Nevertheless, there is a significant and positive effect on ESG after implementing the policy. Thus, similar to Table 6, Figure 3 also provides parallel trend tests, showing that the prerequisite of DID is satisfied.

## 5.2. The Effects of the Green Finance Policy on Individual ESG Pillars

We have seen that the green finance pilot zones policy has a positive and significant effect on firms' ESG activities in total. As the composite ESG score comprises three different pillars, it would be meaningful to examine if the pilot zones policy could also affect each ESG pillar. To answer these questions, we re-estimate the baseline model by replacing the total ESG score with the individual scores. The engagement activities across different ESG dimensions represent various aspects of stakeholder concerns [68]. Chan et al. examined listed firms in the US from 1992 to 2010, and their results showed that firms would make strategic trade-offs between different dimensions [13]. Specifically, financially constrained

firms might first reduce spending on charitable and innovative giving. Yuan et al. subdivided the five CSR dimensions into third-party CSR and stakeholder CSR [63]. Their results showed that more able CEOs invest more in stakeholder-related CSR dimensions. Thus, we consider different dimensions of ESG as fundamental tradeoffs among different stakeholders.

**Table 6.** Dynamic effects of the green finance policy.

| | **(1)** | **(2)** | **(3)** |
|---|---|---|---|
| Treat * year 2014 | −0.204 | −0.156 | −0.182 |
| | (0.915) | (0.916) | (0.911) |
| Treat * year 2015 | 0.234 | 0.265 | 0.252 |
| | (0.872) | (0.875) | (0.873) |
| Treat * year 2016 | 0.192 | 0.246 | 0.230 |
| | (0.875) | (0.878) | (0.875) |
| Treat * year 2017 | 1.090 | 1.147 | 1.143 |
| | (0.783) | (0.786) | (0.785) |
| Treat * year 2018 | 1.598 * | 1.714 ** | 1.684 ** |
| | (0.826) | (0.828) | (0.829) |
| Treat * year 2019 | 1.768 ** | 1.878 ** | 1.825 ** |
| | (0.854) | (0.854) | (0.855) |
| Treat * year 2020 | 1.854 * | 1.921 * | 1.955 * |
| | (0.996) | (0.994) | (1.009) |
| Firm size | 1.066 *** | 1.067 *** | 1.213 *** |
| | (0.193) | (0.201) | (0.214) |
| ROA | 0.314 | 0.524 | 0.567 |
| | (0.549) | (0.575) | (0.567) |
| Duality | 0.106 | 0.0845 | 0.137 |
| | (0.207) | (0.209) | (0.211) |
| Female directors | 0.0716 | 0.126 ** | 0.123 ** |
| | (0.0597) | (0.0612) | (0.0614) |
| Independent directors | −0.547 *** | −0.557 *** | −0.519 *** |
| | (0.168) | (0.174) | (0.176) |
| Largest shareholder | 0.0377 *** | 0.0362 *** | 0.0387 *** |
| | (0.0109) | (0.0111) | (0.0117) |
| SOE | 0.890 ** | 0.887 ** | 0.795 ** |
| | (0.369) | (0.397) | (0.383) |
| Leverage | −0.450 | −0.562 | −0.619 * |
| | (0.318) | (0.347) | (0.370) |
| Growth | −0.0197 | −0.0204 | −0.00953 |
| | (0.0246) | (0.0253) | (0.0276) |
| Free cash flow | 0.0618 | 0.0439 | 0.0415 |
| | (0.0683) | (0.0694) | (0.0702) |
| SA | 21.41 *** | 20.78 *** | 21.15 *** |
| | (2.009) | (2.049) | (2.110) |
| Log(GDP) | | −0.518 | −0.303 |
| | | (0.555) | (0.564) |
| Constant | 78.79 *** | 82.23 *** | 77.16 *** |
| | (8.632) | (10.54) | (10.96) |
| Industry dummies | No | No | Yes |
| Province dummies | No | No | Yes |
| Year dummies | No | No | Yes |
| Firm dummies | Yes | Yes | Yes |
| *N* | 4478 | 4186 | 4186 |
| adj. $R^2$ | 0.795 | 0.798 | 0.799 |

Standard errors in parentheses, * $p < 0.10$, ** $p < 0.05$, *** $p < 0.01$. The standard errors are computed using robust standard errors.

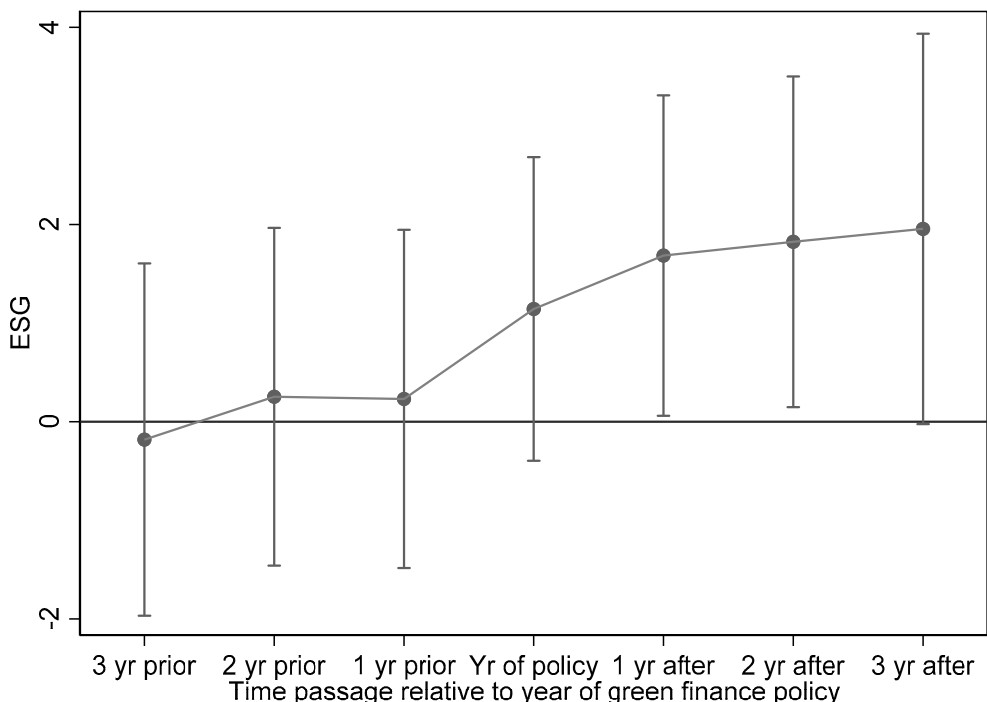

**Figure 2.** The dynamic impact of the green finance policy on the ESG performance.

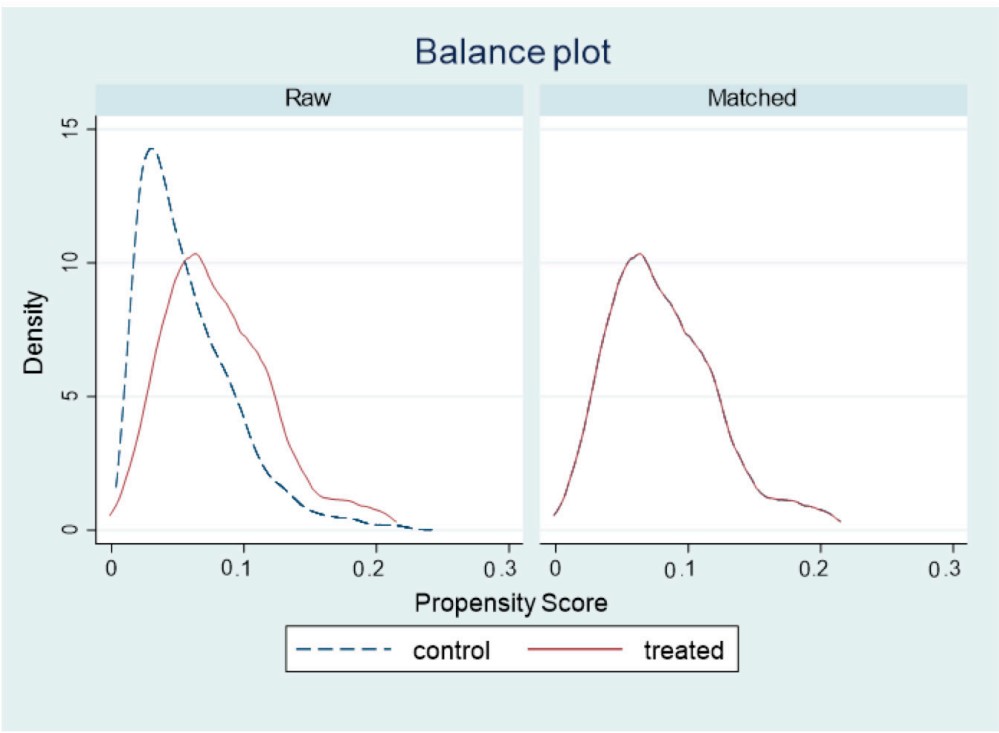

**Figure 3.** Kernel densities pre and post-matching (Nearest-neighbor matching with 0.05 caliper).

As can be seen from Table 7, the overall positive effect of the pilot zones policy on ESG performance is driven by the environmental pillar. For the environmental pillar in column (1), the coefficient of interaction term *treat* ∗ *year* is 1.764 and significant at the 5% level, implying that firms affected by the green finance pilot zones policy significantly enhance their ESG engagement compared to those not influenced by the policy.

**Table 7.** The effects of green finance policy on individual ESG pillars.

|  | E | S | G |
|---|---|---|---|
|  | (1) | (2) | (3) |
| Treat * post | 1.764 ** | 0.918 | 0.314 |
|  | (0.848) | (0.559) | (0.263) |
| Constant | 83.89 *** | 23.12 | 60.89 *** |
|  | (15.63) | (14.34) | (8.722) |
| Control variables | Yes | Yes | Yes |
| Industry dummies | Yes | Yes | Yes |
| Province dummies | Yes | Yes | Yes |
| Year dummies | Yes | Yes | Yes |
| Firm dummies | Yes | Yes | Yes |
| $N$ | 3776 | 4173 | 4186 |
| adj. $R^2$ | 0.732 | 0.771 | 0.809 |

Standard errors in parentheses, * $p < 0.10$, ** $p < 0.05$, *** $p < 0.01$. The standard errors are computed using robust standard errors.

Notably, the coefficients of the interaction are insignificant in columns (2) and (3), indicating that the green finance policy fails to have a significant impact on the social and governance pillars. This is easy to understand since the green finance policy is designed to be effective in environmental aspects. For example, Zhang et al. examined the same green finance pilot policy, and they showed that the green finance policy is effective in air pollution control, which is an essential aspect of the environmental pillar of ESG [7]. Using provincial panel data during 2011–2019, Huang and Zhang examined the same policy, and they found that macroeconomically, the green finance pilot policy plays a vital role in pollution control and is conducive to environmental enhancement.

*5.3. Heterogeneity*

The average effect of the green finance policy on firms' ESG performance was demonstrated above. Our sample covers various firms, and there are also differences in firms' ESG performance within industries. It is interesting, therefore, to explore the asymmetric effects of the policy. Yu et al. found that generally, green finance policies could effectively mitigate financing restraints on green innovation, but POEs are less likely to obtain green credits. Their results imply that POEs are in a disadvantaged position to get credits under the existing green finance system [4]. Moreover, Hong et al. showed that firms with fewer financial constraints are more likely to invest in CSR [18]. In addition, Xu and Li showed that the green credit policy has asymmetric impacts on "two high" firms and green firms [78]. Specifically, the green credit policy increases the debt financing cost of "two-high" firms but decreases that of green firms. Thus, as in other green finance policy studies [16,20,28,38,94], we also discussed the heterogeneity in this section. We use both subsample regression [16,20,94] and triple differences methods [28,38].

Table 8 shows the heterogenous effects of the pilot policy for non-SOEs versus SOEs, high financial constrained versus low financial constrained, and non-heavy-polluting versus heavy-polluting firms in columns (1)–(6), respectively.

In China, tighter connection with the government possibly grants SOEs better access to external financing, while this may not be the case for non-SOEs [38]. Therefore, we predict stronger impacts of the pilot policy on SOEs. As presented in columns (1) and (2) of Table 8, consistent with Zhang et al., we also find that the policy has a larger effect among state-owned firms [92]. Compared to SOEs, non-SOEs tend to face higher financing constraints due to uneven loan availability from state-owned banks [95]. If financial distress accounts for firms' poor environmental performance, the inferior environmental performance of non-SOEs implies that non-SOEs have limited access to financing resources. Jin et al. showed that banks are still the main provider of credit in the Chinese financial market [96]. This result is also consistent with the literature on the relationship between financial constraints and pollution control. Zhang et al. found that SOEs mainly resort to external financing

resources to reduce emissions, while POEs mainly financed internally [97]. To further examine the heterogenous effect of SOEs, we substitute the triple interaction in equation (3) with the triple interaction term *treat \* post \* SOE*. As shown in column (4) of Table 9, the effect of green finance on ESG performance is significantly positive for SOEs relative to non-SOEs.

**Table 8.** Heterogeneity.

|  | Non-SOEs | SOEs | High Financial Constrained Firms | Low Financial Constrained Firms | Non-Heavy-Polluting Firms | Heavy-Polluting Firms |
|---|---|---|---|---|---|---|
|  | (1) | (2) | (3) | (4) | (5) | (6) |
| Treat * post | 1.106 ** | 2.255 *** | 0.654 | 3.356 *** | 1.403 ** | 1.299 *** |
|  | (0.438) | (0.682) | (0.784) | (0.814) | (0.680) | (0.491) |
| Constant | 58.21 *** | 38.44 ** | 20.90 | 86.67 *** | 81.06 *** | 26.34 * |
|  | (15.05) | (14.95) | (17.17) | (21.01) | (14.15) | (15.09) |
| Control variables | Yes | Yes | Yes | Yes | Yes | Yes |
| Industry dummies | Yes | Yes | Yes | Yes | Yes | Yes |
| Province dummies | Yes | Yes | Yes | Yes | Yes | Yes |
| Year dummies | Yes | Yes | Yes | Yes | Yes | Yes |
| Firm dummies | Yes | Yes | Yes | Yes | Yes | Yes |
| $N$ | 2276 | 1895 | 2067 | 2061 | 2123 | 2056 |
| adj. $R^2$ | 0.809 | 0.784 | 0.812 | 0.816 | 0.817 | 0.779 |

Standard errors in parentheses, * $p < 0.10$, ** $p < 0.05$, *** $p < 0.01$. The standard errors are computed using robust standard errors.

**Table 9.** Triple difference test of Heterogenous effects.

|  | (1) | (2) | (3) | (4) |
|---|---|---|---|---|
| Treat * post * Pollution | −0.00844 |  |  |  |
|  | (0.839) |  |  |  |
| Treat * post * highGDP |  | 1.701 ** |  |  |
|  |  | (0.819) |  |  |
| Treat * post * highSA |  |  | 2.746 *** |  |
|  |  |  | (1.042) |  |
| Treat * post * SOE |  |  |  | 1.698 ** |
|  |  |  |  | (0.785) |
| Constant | 76.08 *** | 74.14 *** | 72.85 *** | 75.20 *** |
|  | (10.79) | (10.94) | (10.99) | (11.01) |
| Control variables | Yes | Yes | Yes | Yes |
| Industry dummies | Yes | Yes | Yes | Yes |
| Province dummies | Yes | Yes | Yes | Yes |
| Year dummies | Yes | Yes | Yes | Yes |
| Firm dummies | Yes | Yes | Yes | Yes |
| $N$ | 4186 | 4161 | 4186 | 4186 |
| adj. $R^2$ | 0.799 | 0.799 | 0.800 | 0.799 |

Standard errors in parentheses, * $p < 0.10$, ** $p < 0.05$, *** $p < 0.01$. The standard errors are computed using robust standard errors.

Financial constraint serves as a key mechanism through which green finance influences firms' environmental performance. Thus, the green finance policy might have heterogenous effects on firms with different levels of financial constraints. The results in columns (3) and (4) of Table 8 show that the green finance policy has a more significant effect on firms with low financial constraints. Meanwhile, the green finance policy has no significant effect on

firms with high financial constraints. This implies that firms in a weak financial situation are less likely to engage in CSR. Consistent with our findings, Chan et al. suggested a significant negative relationship between financial constraints and CSR engagement [13]. To further examine the heterogenous effect of financial constraints, we substitute the triple interaction in Equation (3) with the triple interaction term $treat * post * highSA$, where $highSA$ is a dummy variable equal to 1 for firms with SA index above the sample median (low financial constraints). As shown in column (3) of Table 9, the effect of green finance on ESG performance is significantly positive for less financially constrained firms than firms with high financial constraints.

Environmental damage is mainly caused by heavy-polluting firms. Thus, it is important to assess whether green finance policy could promote ESG activities of heavy-polluting firms. Results in columns (5) and (6) of Table 8 show that the green finance policy plays an important role in both non-heavy-polluting and heavy-polluting firms. To further investigate the effect of green finance policy on firms' ESG performance varied by pollution level, we use a triple interaction term as specified in Equation (3), as in previous literature [38]. As shown in column (1) of Table 9, the effect of the green finance policy on firms' ESG performance is insignificant for heavy-polluting firms relative to non-heavy-polluting firms, indicating that there's no significant heterogeneity between heavy-polluting and non-heavy-polluting firms.

As indicated in Table 1, the green finance pilot zones policy has different priorities in different regions. Financial support is largely determined by the level of local economic development. For example, in Zhejiang and Jiangsu, which are more economically developed, more advanced instruments, such as green bonds and green credit systems, are stressed. In Xinjiang, the priority is to optimize local ecological resources. We use city-level GDP per capita to measure the level of economic development. To examine the effect of heterogenous green finance policy in different pilot zones, we substitute the triple interaction term in Equation (3) with the triple interaction term $treat * post * highGDP$, where $highGDP$ is a dummy variable equal to 1 for firms in pilot zones with city-level GDP per capita above the sample median and 0 otherwise. As shown in column (2) of Table 9, the effect of green finance policy on firms' ESG performance is significantly positive for more developed zones relative to less developed areas. Consequently, the ESG performance of firms in more developed zones improves more when exposed to the pilot policy.

*5.4. Mechanism Analysis*

Based on the above analysis, we have shown that the green finance pilot policy has a positive and significant effect on firms' ESG engagement. As we mentioned before, previous studies have pointed out that financial constraints are a key driver of corporate social responsibility [18,36]. To show the role of financial constraints in the relationship between the green finance pilot zones policy and firms' ESG performance, we substitute the dependent variable with the SA index and re-estimate Equation (1).

Column (1) of Table 10 presents the results. Consistent with Li et al., the pilot policy has a negative and significant effect on the SA index, indicating that the pilot policy tightens firms' financial constraints [28]. They investigated the 2012 China's Green Credit Guidelines using listed firms from 2009–2020, and they found that the green credit policy could worsen firms' financial constraints. Moreover, our result is consistent with Liu et al., which utilized a financial computable general equilibrium model (CGE) and found that with the implementation of the green credit policy, the average financing costs of the energy-intensive industries increase by 1.1% in the short run, decrease by 0.008% in the medium run, and increase by 0.003% in the long run [77]. The pilot zones policy was implemented in June 2017, and our sample period is 2013–2020. Thus, we might only observe the short-run effect of the policy. The higher financing costs further tighten financial constraints in the short run. Despite the differences in priorities in different pilot zones, the pilot zones all use capital allocation to steer financial resources into green projects and technologies.

Thus, heavy-polluting firms could easily suffer from higher thresholds and capital costs in external financing [37].

**Table 10.** Mechanism analysis: financial constraints.

| | Full Sample | High FC | Low FC | Non-SOEs | SOEs | Non-Heavy-Polluting Firms | Heavy-Polluting Firms |
|---|---|---|---|---|---|---|---|
| | (1) | (2) | (3) | (4) | (5) | (6) | (7) |
| Treat * post | −0.0140 *** | 0.00495 | −0.0150 | −0.0315 *** | −0.000647 | −0.00517 | −0.0183 *** |
| | (0.00498) | (0.00719) | (0.0112) | (0.00588) | (0.00774) | (0.00687) | (0.00691) |
| Constant | −3.734 *** | −3.966 *** | −3.626 *** | −3.355 *** | −3.864 *** | −3.568 *** | −3.746 *** |
| | (0.0788) | (0.0731) | (0.188) | (0.116) | (0.105) | (0.130) | (0.0975) |
| Control variables | Yes | Yes | Yes | Yes | Yes | Yes | Yes |
| Industry dummies | Yes | Yes | Yes | Yes | Yes | Yes | Yes |
| Province dummies | Yes | Yes | Yes | Yes | Yes | Yes | Yes |
| Year dummies | Yes | Yes | Yes | Yes | Yes | Yes | Yes |
| Firm dummies | Yes | Yes | Yes | Yes | Yes | Yes | Yes |
| $N$ | 4186 | 2067 | 2061 | 2276 | 1895 | 2123 | 2056 |
| adj. $R^2$ | 0.975 | 0.968 | 0.932 | 0.976 | 0.974 | 0.978 | 0.970 |

Standard errors in parentheses, * $p < 0.10$, ** $p < 0.05$, *** $p < 0.01$. The standard errors are computed using robust standard errors. Firm age and firm size are not included as control variables.

To further analyze the heterogenous effect on financial constraints, we re-estimate the same specification as in column (1) of Table 10 but using the subsamples: firms with high financial constraints (with the SA index lower than the sample median), firms with low financial constraints, SOEs, non-SOEs, non-heavy-polluting firms, and heavy-polluting firms for columns (2)–(7), respectively. The results show that the pilot policy tightens financial constraints significantly in non-SOEs and heavy-polluting firms but has no significant effect in both highly and low-constrained firms, SOEs, and non-heavy-polluting firms.

Unsurprisingly, the heavy-polluting firms become more financially constrained since the main aim of the green finance policy is to support environment-friendly projects. Our results indicate that the pilot policy has worsened the financial constraints of less constrained firms more relative to high-constrained firms, though insignificantly. This might be due to the sample composition. For example, there are 56.1% of non-SOEs in less constrained firms, while 53.4% in highly constrained firms. Non-SOEs are more likely to be financially constrained because of the pilot policy, as shown in columns (4) and (5). Yu et al. showed that green finance policies primarily mitigated the financing constraints of SOEs, and POEs are less likely to obtain green credits [4].

In sum, the mechanism analysis reveals that the pilot policy promotes firms' ESG performance even if it worsens their overall financial constraints. The reason might be that the policy aims at environment-friendly projects. Moreover, the effects on financial constraints are heterogenous, with less financially constrained firms, non-SOEs, and heavy-polluting firms experiencing a statistically significant increase in financial constraints when exposed to the pilot zones policy.

## 6. Robustness Tests

### 6.1. PSM-DID

The pilot policy is implemented in only eight cities, and therefore only a relatively small share of firms are exposed to the policy. Thus, there may exist a small sample selection bias [37]. To mitigate the potential self-selection bias, as in other studies [16,20,25,28,38], we utilize the propensity score matching (PSM) method to match firms in pilot cities and firms not exposed to the policy. The key idea of PSM is to construct a counterfactual control group to investigate the effect of the changes on ESG performance. We use one-to-three nearest-neighbor matching with 0.05 caliper, one-to-three nearest-neighbor matching

with 0.001 caliper, radius matching with 0.001 radius, and kernel matching to show the consistency of our results.

To demonstrate the difference in pre and post-matching, Figure 3 presents the kernel densities of scores for the two groups. The left part depicts kernel densities for the treatment and control groups before matching, while the right part presents kernel densities post-matching (one-to-three nearest-neighbor matching with 0.05 caliper). The distributions after matching are more consistent and almost overlap with each other.

Table 11 shows the PSM-DID results. They are quite similar to the baseline regression findings. That is, the net effect of the green finance policy on firms' ESG performance is still positive and significant, implying that our baseline results are not biased by unobserved factors.

**Table 11.** PSM-DID results.

| | Nearest (c = 0.05) | Nearest (c = 0.001) | Radius | Kernel |
|---|---|---|---|---|
| | (1) | (2) | (3) | (4) |
| Treat * post | 1.601 *** | 1.667 *** | 1.667 *** | 1.601 *** |
| | (0.396) | (0.403) | (0.403) | (0.396) |
| Constant | 62.69 *** | 59.46 *** | 59.46 *** | 62.69 *** |
| | (11.88) | (11.98) | (11.98) | (11.88) |
| Control variables | Yes | Yes | Yes | Yes |
| Industry dummies | Yes | Yes | Yes | Yes |
| Province dummies | Yes | Yes | Yes | Yes |
| Year dummies | Yes | Yes | Yes | Yes |
| Firm dummies | Yes | Yes | Yes | Yes |
| $N$ | 3920 | 3761 | 3761 | 3920 |
| adj. $R^2$ | 0.798 | 0.801 | 0.801 | 0.798 |

Standard errors in parentheses, * $p < 0.10$, ** $p < 0.05$, *** $p < 0.01$. The standard errors are computed using robust standard errors. The propensity score is estimated using logit model. Province dummies, industry dummies, and year dummies are controlled.

### 6.2. Alternative Proxy for ESG Performance

Similar to other ESG studies, the ESG scores might suffer from reporting bias. Only about 30% of listed firms have Bloomberg ESG ratings, due to inadequate ESG disclosure information [85]. Moreover, Sun and Saat mentioned that Bloomberg only covers about 1000 listed firms in China [61]. To rule out the possibility that the coverage of the ESG score might bias the results, we also use Runling CSR to show the robustness of our results. Runling ESG is retrieved from the CSR rating agency Runling (also known as RKS). RKS organization and the Runling CSR scores for Chinese firms are modeled after the US CSR rating agency KLD. Thus, similar to studies based on KLD CSR scores [52,98], the Runling CSR score is also widely used as a composite measure of CSR for Chinese firms in the literature [99–102]. Table 12 shows the results using Runling CSR ratings as the alternative proxy for ESG performance. The results presented in Table 12 are consistent with the baseline results.

### 6.3. Placebo Test

To examine if omitted variables bias the results, as in other studies [6,7,15,19,97], a placebo test is performed by randomly assigning the implementation of the pilot policy to cities [103]. Eight cities are randomly selected at time $t$, which are selected at random between 2014 and 2018, allowing for at least one year before and one year after the policy to implement the DID method. Using this false green finance status variable and time variable, we conduct a placebo estimation using the same specification in column (5) of Table 5. The false green finance policy variable should have generated an insignificant estimate with a

magnitude near zero; otherwise, it would suggest a misspecification of the DID estimation. It is repeated 500 times to enhance the identification power of this placebo test.

**Table 12.** Runling ESG.

|  | (1) | (2) | (3) | (4) |
|---|---|---|---|---|
| Treat * post | 2.708 ** | 2.709 ** | 2.826 ** | 2.894 ** |
|  | (1.267) | (1.267) | (1.328) | (1.309) |
| Constant | −12.38 | −13.39 | −13.28 | −3.926 |
|  | (19.71) | (19.79) | (19.87) | (20.81) |
| Control variables | Yes | Yes | Yes | Yes |
| Industry dummies | No | No | Yes | Yes |
| Province dummies | No | Yes | Yes | Yes |
| Year dummies | No | No | No | Yes |
| Firm dummies | Yes | Yes | Yes | Yes |
| $N$ | 2060 | 2060 | 2060 | 2060 |
| adj. $R^2$ | 0.834 | 0.833 | 0.833 | 0.837 |

Standard errors in parentheses, * $p < 0.10$, ** $p < 0.05$, *** $p < 0.01$. The standard errors are computed using robust standard errors.

Figure 4 presents the distribution of the estimates from the 500 runs. We can see that the observations are scattered and resemble a symmetric bell-like shape. Moreover, the *p*-value of the Jarque–Bera test of the estimates is 0.1694. Thus, we fail to reject the null hypothesis of the normality of the distributions of the estimates. The estimates are normally distributed and scattered around 0, indicating that an artificial "counterfeit variable" has no effect on firms' ESG performance. The results imply that it is the green finance pilot zones policy rather than other policies in pilot zones that have led to the increase in firms' ESG performance.

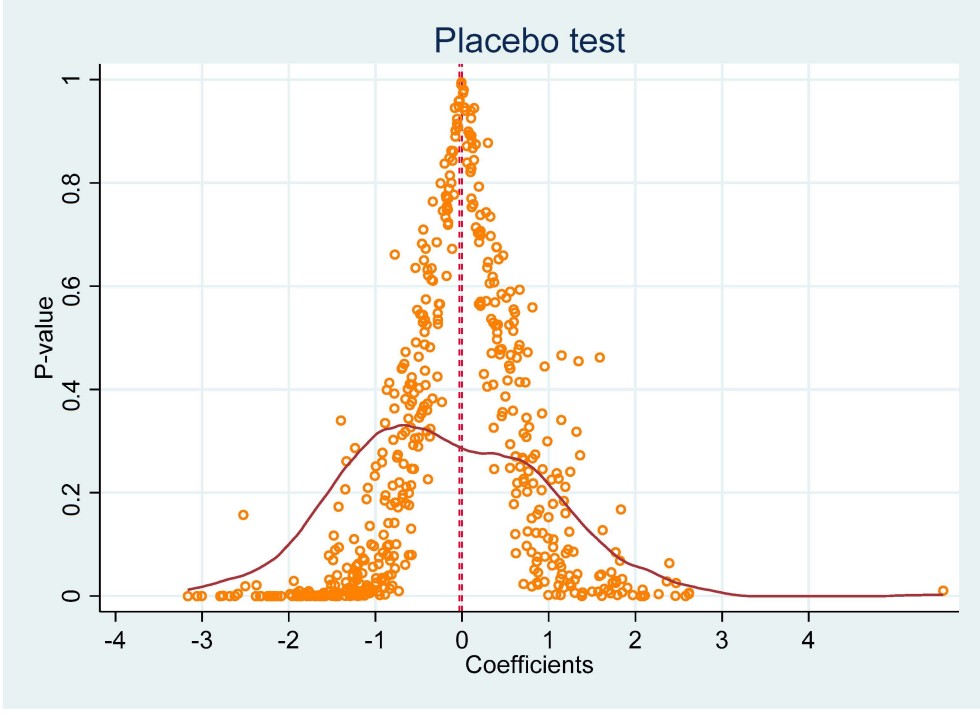

**Figure 4.** Distribution of estimated coefficients of falsification test. The figure shows the cumulative distribution density of the estimated coefficients is from 500 simulations randomly assigning the green finance policy status to cities.

*6.4. Sustainability Implications and Policy Recommendations*

Our results have important sustainability implications. Firstly, green finance policy could be a powerful instrument to enhance firms' sustainability. Green finance is designed to make sustainability a part of financial decision making. Our results indicate that green finance policy is effective in fostering firms' sustainability, measured by ESG ratings. Moreover, better ESG performance is usually related to higher firm value, less risk, and better access to finance [54–57].

Secondly, better ESG performance signals firms' endeavor in reducing pollution and improving energy efficiency. In particular, our results indicate that firms' ESG growth has been mainly driven by the environmental pillar. Thus, the improvement in firms' ESG performance induced by green finance policy is vital for environmental protection, climate change mitigation, and thus for society's sustainability.

Thirdly, different from other studies, our results imply that green finance policy could promote sustainability through sustainable investment. Socially responsible investors rely heavily on the ESG scores [11]. However, ESG investing has yet to become mainstream in China. Globally, around 33% of all professionally managed assets are subject to ESG criteria [104]. The number of global investor signatories to the United Nations Principles for Responsible Investment (UN PRI) topped 3000 in 2020, up from 63 in 2006. Total assets under management (AUM) represented by the signatories exceeded USD 100 trillion in 2020 [105]. According to PwC's prediction, ESG-oriented AUM in the US would increase to 10.5 trillion dollars in 2026 from 4.5 trillion US dollars in 2021; in Europe, it would grow 53% to 19.6 trillion dollars. The Asia-Pacific is expected to more than triple, reaching 3.3 trillion dollars [106]. In China, except for money market funds, total ESG exchange-traded funds (ETF) and mutual funds experienced AUM more than triple in 2020 to reach 28.5 billion dollars, with net inflows reaching 10.5 billion dollars [107].

Our results indicate that the green finance policy plays a vital role in fostering firms' ESG, which is the basis of sustainable investing. Thus, our results imply that the green finance policy could be supportive of sustainable investment in China. Pástor et al. suggested that sustainable investing exerts its social impacts in two ways [108]. First, it promotes firms to become greener. Second, it encourages green firms to invest more and decreases investment by brown firms. Therefore, our results also have a vital implication for developing a greener society at large.

The results have important policy implications. ESG increases a firm's value and also serves as a vital driving force to promote the sustainable development of firms. However, motivating firms to engage more in ESG in emerging countries like China is still challenging. Our results imply that green finance could be an effective instrument to improve firms' ESG performance. Moreover, while vigorously promoting green finance, the policymaker should also allocate more financial resources to non-SOEs which are usually less financially constrained.

## 7. Conclusions

This paper explores the 2017 green finance pilot zones policy as a quasi-natural experiment to examine the effects of the green finance policy on firms' ESG performance. The findings show that, firstly, the 2017 green finance pilot zones policy has had a significant and positive effect on firms' ESG performances. Secondly, the overall positive effect of the green finance policy on ESG performance is driven mainly by the environmental pillar rather than the social and governance pillars.

Thirdly, utilizing the subsample estimation and triple differences method, we further find that the higher ESG performance is driven mainly by heavy-polluting firms, firms with less financial constraints, firms in economically more developed zones, and SOEs. Fourthly, the mechanism analysis reveals that the pilot policy improves firms' ESG performance even if it worsens their overall financial constraints. The reason might be that the policy aims at environment-friendly projects. Moreover, the effects on financial constraints are heterogenous, with less financially constrained firms, non-SOEs, and heavy-polluting firms

experiencing a statistically significant increase in financial constraints when exposed to the pilot zones policy. Finally, the results are robust to the parallel trend test, PSM-DID, alternative ESG proxy, and placebo test.

Previous studies have shown that green finance policy could foster firms' green performance and green innovation [20,23]. Consistent with their findings, our results suggest that green finance policy could enhance the environmental pillar of a firm's ESG. Moreover, different from their conclusions, we find that the green finance policy has a positive and significant effect on firms' ESG performance. Socially responsible investors rely heavily on the ESG scores [11]. Therefore, our results have important implications for establishing a better green finance system in China.

Moreover, heterogeneity analysis shows that the higher ESG performance is mainly driven by firms with less financial constraints, firms in economically more developed areas, and SOEs. This confirms that firms with fewer financial constraints are more likely to engage in ESG [13,18].

In addition, consistent with Li et al. [28], the mechanism analysis reveals that the pilot policy fosters firms' ESG performance even if it worsens their overall financial constraints. The reason might be that the policy aims at environment-friendly projects. Moreover, our results show that the tightening financial impacts are asymmetric among different types of firms. Specifically, the green finance pilot policy worsens the financial constraints of non-SOE firms and high-polluting firms more. Our results confirm the findings of Yu et al. and Xu and Li [4,78].

However, this paper has the following limitations. Firstly, due to insufficient ESG disclosure information, our sample only includes listed firms, which usually outperform non-listed firms in size and profitability. Thus, our conclusion cannot be directly extrapolated to non-listed firms. In the future, when there are more data on ESG performance, the research could extend to a wider range of firms. Secondly, our results indicate that the pilot zones policy worsens firms' overall financial constraints. In the literature, Li et al. found that green finance policy increases firms' overall financial constraints [28], but Yu et al. showed that green finance policies could effectively ease financial constraints on green finance [4]. The reason might be that the pilot zones policy was first implemented in June 2017, and our sample period is 2013–2020. Thus, we might only be observing the short-run effect of the policy. We look forward to further research with an expanded sample period to examine the long-term effect of the pilot zones policy on firms' financial constraints. Thirdly, previous studies have shown that financial constraint is an essential channel through which green finance impacts firms' ESG performance. However, as we have no access to detailed firm-level or regional-level green credit data, we could not examine the underlying mechanisms in depth. In the future, more detailed information on green credit, green insurance, green bonds, and green security could help us to dig deep into the underlying mechanisms of the green finance policy, and the corresponding conclusions should help us to better understand the effects of green finance.

**Author Contributions:** Conceptualization, X.S. and Z.G.; methodology, X.S. and Z.G.; software, X.S. and C.Z.; validation, X.S., C.Z. and Z.G.; formal analysis, X.S.; investigation, X.S., C.Z. and Z.G.; resources, X.S.; data curation, X.S.; writing—original draft preparation, X.S.; writing—review and editing, Z.G.; visualization, C.Z.; supervision, X.S.; project administration, X.S.; funding acquisition, X.S. All authors have read and agreed to the published version of the manuscript.

**Funding:** This research was funded by the National Natural Science Foundation of China (Grant #72103167).

**Institutional Review Board Statement:** Not applicable.

**Informed Consent Statement:** Not applicable.

**Data Availability Statement:** Not applicable.

**Acknowledgments:** We would like to thank the editor and anonymous referees for their constructive comments that helped to improve the paper.

**Conflicts of Interest:** The authors declare no conflict of interest.

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
