# Peer review of "Green Finance Policy and ESG Performance: Evidence from Chinese Manufacturing Firms"

_sustainability, doi:10.3390/su15086781_

Round 1

Reviewer 1 Report

The paper is interesting and it has merits for the readers in the sustainability field. Further, green finance and ESG are two new fields to be investigated with respect to their relations. The topic is a good choice for research. I congradulate authors for this respect. 

The paper has certain issues that need to be addressed. See the critiques below. 

1) Percentage of similarity in similarity software yielded a quite high percentage for a research paper.  13% and 12% are from the sources below. The first is I assume another study of the authors. The similarity should be reduced. 

1. https://www.mdpi.com/1660-4601/19/18/11287/htm

2. https://www.mdpi.com/2071-1050/15/2/1619

2) Abstract: Data and method are not discussed in the abstract. It focuses directly to the results. It is not clear what is studied. The contribution is not highlighted. What is the novelty of sample? What contribution does the method yield compared to the existend methods? The abstract should be rewritten to emphasize contribution. 

3)  At line 62, "However, up to now, few studies have explored 62 the combined economic, social, and environmental effects of green finance at the firm 63 level." This sentence needs to be maintained by discussing these few studies and highlighting the difference of your paper in terms of novelty. 

4) What is meant in this sentence is not clear. "Moreover, our findings also inform work studying the role of green finance policy 106 on sustainable development." Revise please. 

5) Many specific information are given on China in the institutional background section. However, no references are used in many places for specific information. Just few sources are cited in the whole section. Most probably, statistical agencies and statistical reports should be cited in this section. Revise please. 

6) In the literature review, methods, data are omitted in generally with minor exceptions. They should be given in the text. With which methods and data types are these findings found? For example at line 257, which method is used, I assume a cointegration method, which cointegration method, if so to obtain long-run and short-run results?  

7)  What does the sentence at line 273 mean? It has grammatical errors and other than that, it is not clear. 

8) At line 269, the assumption that these areas are "exogeneous shock" in DID setting is too harsh. "An exogeneously applied policy" that leads to a DID investigation, before and after.. is more appropriate. 

9) Data is explained clearly in data section. Sample size is adequate. It is better to add the sources as annotations to Table 2 as a column. I also suggest adding JB test statistic as a column. Also, units are not certain for some variables. make sure that they exist in text. Example, free cash flow.

10) Why are some variables are in logarithms while others are not? An explanation should be made to choose lin-log form for these variables in the models. 

11) Line 420: in the comment for parameter of interaction, first the treat and post effects should be given and through significance of interaction, the combined effect accelerates. It should be added. Only the interaction is noted. 

12)  Heterogeneity in Table 7 should be accompanied with heterogeneity tests if possible. (This comment is optional)

13) Insignificance of parameters in the High financial constrained firms model should be discussed more. Why are they insignificant, what would be the reason? It is not stated at line 530. 

14) At line 670, "Figure 4 presents the distribution of the estimates from the 500 runs. We could see 670 that the estimates are normally distributed", the sentence is not statistically tested, normality cannot be stated by looking at the figure only. However, either the sentence should be revised to "the observations are scattered that resembles a symmetric bell like shape." also, normality test result could be added to the sentence. 

14) Policy implications, future directions are adequate in the and in line with findings in the conclusion section.  Since the paper is submitted to sustainability, it is advised to include a final section before conclusion. Sub heading suggestion: "Sustainability Implications and Policy Recommendations". Bring policy recommendations from conclusion to this section. However, sustainability is not discussed in the paper. Discuss accordingly. Discuss societal impacts, environmental impacts and possible implications to climate change. The contribution of the paper's findings should be discussed both in this section and in the conclusion section.    

My overall evaluation is, the paper is an important paper after corrections. It will greatly improve, especially in terms of its contribution being not highlighted in the paper. But first, the high percentage of similarity should be attended immediately by the authors. It is high for a research paper, over 30%. Reduction to less than 20 and more preferably, 15% is needed for a research paper. Self citations and citations to papers are recommended.  

My decision is major revision.  

Author Response

Dear reviewer:

We want to express our gratitude for your thoughtful suggestions and most useful comments, which we have made every attempt to take into account in a revised version of the paper. We also have carefully addressed all comments in the attached file.

I also include the plagiarism report from Turnitin for you to check. 

We really appreciate your comments and suggestions. If there's any other questions, please let me know.

Thank you very much.

Best,

Xiuli

Reviewer 2 Report

1.              The paper's abstract must be improved according to why, how, and for whom this study was conducted. Do not use any abbreviation when it first occurs, such as PSM and DID in the Abstract. Please add its full name when the abbreviation occurs. A word between 150-200 could be sufficient in this regard.

2.              The introduction is well written; however, the motivation and the contribution need to be further strengthened. This will increase the value of the paper for readers.

3.              The authors should take into account the most recent information regarding the focal variable in the literature. To enhance the discussion in the literature, please include earlier studies in 2020 and 2022.

4.              In the conclusion part, the authors should list all the limitations and discuss possible directions for further research.

5.              Modify the paper based on the journal’s guidelines, especially the references.

6.              Please check the manuscript again for errors.

Author Response

Dear reviewer:

We want to express our gratitude for your thoughtful suggestions and most useful comments, which we have made every attempt to take into account in a revised version of the paper. We also have carefully addressed all comments in the attached file.

We really appreciate your comments and suggestions. If there's any other questions, please let me know.

Thank you very much.

Best,

Xiuli

Reviewer 3 Report

This paper uses data of Chinese listed companies from 2013 to 2020 to study the effects of the green finance pilot zones policy on ESG permanence. I find the topic interesting and the paper well-written. I only have a few minor suggestions.

1. The main specification of the paper controls firm fixed effects and year fixed effects. There is no need to add the two variables Treat and Post in equation 1 (and Treat in equation 2) because the two variables are automatically absorbed by the fixed effects. Table 4 does not need to report the coefficients of Post for the same reason.

2. The triple differences (equation 3) should consider adding the interaction of Treat and Post and the interaction of Pollution and Post, otherwise the coefficient of the triple interaction will be difficult to explain since multiple groups would be used as the reference. So does the analyses in Table 8.

3. The measure of financial constraints is determined by size and age. Since size and firm fixed effects are controlled in Table 9, what variations are left in the dependent variable and why does the left variations matter for financial constraints?

4. A typo in the title of Figure 4: "The figure shows he cumulative distribution density".

Author Response

(The authors gave the same response as above.)

Reviewer 4 Report

Introduction: Prior to delving into the topic of ESG, it is imperative to underscore its significance in the present era, particularly in light of pressing concerns such as climate change and extreme weather events, which pose existential threats to the planet. Through implementing environmentally conscious measures, businesses can play a pivotal role in safeguarding the ecosystem and biodiversity of a given region or state.

Also extend the introduction by highlighting why it is important to promote green financing in China, for example, given the scale of environmental challenges faced by China, including air and water pollution, soil contamination, and climate change, it is imperative that businesses adopt sustainable practices and transition to a low-carbon economy. By improving their green financing, firms can secure funding for projects that promote environmental protection, resource efficiency, and clean energy, thereby contributing to China's sustainable development goals and mitigating the negative impacts of economic growth.

I urge the authors to extend the introduction on the above lines.

Why the study was conducted in manufacturing? The authors say, In 2012, the manufacturing sector accounted for 31.11% 69 of total carbon emissions and 56.86% of total energy consumption in China.

However, this reference is more than a decade ago. Please provide up to date justifications to conduct this study in the manufacturing sector.

Literature: what was the criteria to identify the selected factors as important ones to predict ESG?

Estimation strategy:

While the difference-in-differences (DID) method is a widely used statistical technique for estimating causal effects, there are several potential critiques which I want the authors to explain carefully, how they dealt with these issue?

  For example, DID relies on the assumption of parallel trends, which means that the treatment and control groups would have had similar outcomes in the absence of the treatment. However, if there are other factors that affect the outcome differently in the treatment and control groups over time, then the assumption of parallel trends may not hold. This could lead to biased estimates of the treatment effect. Another issue is that the DID method is sensitive to the choice of the control group. If the control group is not well-matched to the treatment group, then the estimated treatment effect may be biased. Additionally, if the control group experiences changes that are not captured by the researcher, then the estimated treatment effect may be confounded. Moreover, the DID method assumes that the treatment is "on" or "off" at a specific point in time, which may not reflect the true nature of the intervention. In some cases, the treatment may be gradual or ongoing, and the DID method may not capture these nuances.

Discussion: Please discuss you findings by comparing previous research. Provide some important theoretical and practical implications for the readers.

Also highlight the limitations of your study.

Good Luck 

Author Response

(The authors gave the same response as above.)

Reviewer 5 Report

The topics of this paper are interesting. The structure and content must be revised, and authors must better explain results before being reconsidered for publication.

The introduction has to clarify the research questions of this study better and provide more theoretical background about the concept of Green Finance Policy and ESG Performance. Authors have to describe better the different sources of Green Finance Policy. 

Section 2 can be called "literature review". A table representing the established studies in ESG Performance with their used theories and country of studies can present a better overview of the current paper.

Results. This section seems well developed and the results presented clearly. Please elaborate on this method and provide some related references for this technique in the related sections. Also, please report some model fit indices.

The discussion "conclusion" has to compare these results with previous studies to show the real breakthrough of this study.

Why is the model important?

The theoretical and managerial implications must be here clearly discussed and underpinned by results and literature. The current format is too weak and can not be acceptable for publication in the journal.

Please look at the reference list closely as there are some minor errors.

Author Response

(The authors gave the same response as above.)

Round 2

Reviewer 1 Report

The responses of the authors are in great detail and there is great efford given to improve the paper. Thank you for taking my comments into careful consideration.

My decision for this final version of the paper is to accept. 

Congrats to the authors. 

Author Response

Dear referee,

We are grateful for your valuable suggestions and comments. They greatly improve the quality of the manuscript. Thank you very much!

Sincerely,

Xiuli

Reviewer 4 Report

The authors have addressed most of my concerns. 

Author Response

(The authors gave the same response as above.)
